**www.cambridge.org/qrd**

## Perspective

structural biology; protein complex; protein-protein interactions; cross-linking mass spectrometry; integrative structural modelling

**Corresponding author:**
Zheng Ser;
Email: ser_zheng@imcb.a-star.edu.sg

# Filling in missing puzzle pieces in protein structural biology with cross-linking mass spectrometry

Zheng Ser 

Institute of Molecular and Cell Biology (IMCB), Agency for Science, Technology and Research (A*STAR), Singapore 138673, Singapore

## Abstract

High resolution structures of protein complexes provide a wealth of information on protein structure and function. Databases of these protein structures are also used for artificial-intelligence (AI)-based methods of structural modelling. Despite the wealth of protein structures that have been determined by structural biologists, there are still gaps, or missing pieces in the puzzle of protein structural biology. Highly flexible regions may be missing from protein structures and conformational changes of different protein complex states may not be captured by current databases. In this perspective, I sketch out several ways that cross-linking mass spectrometry can contribute to filling in some of these missing pieces: Identification of cross-linked interactions in highly flexible protein regions not captured by other structural techniques; capturing conformational changes of protein complexes in different functional states; serving as distance constraints in integrative structural modelling and providing structural information of *in cellulo* proteins. The myriad ways in which cross-linking mass spectrometry contributes to filling in missing pieces in structural biology makes it a powerful technique in structural biology.

High-resolution protein structures enable biologists to understand protein functions down to the atomic level. A large number of protein structures have been solved and deposited on protein structure databases from a variety of techniques such as X-ray crystallography, cryo-electron microscope (cryo-EM), and nuclear magnetic resonance (NMR). These structures may contain artifacts from the associated techniques such as compaction from crystallization, and modifications or truncations to protein sequences which result in higher-resolution structures (Niedzialkowska *et al.*, 2016). Additionally, protein regions that are flexible or highly mobile are typically not captured by X-ray crystallography and cryo-EM. These missing structural regions or missing pieces may play important roles in protein structure, interaction, and function. Additionally, artificial intelligence (AI)-based methods for structural modelling draw upon current protein structure databases and may propagate artifacts in these structures to predicted structures. Here, in this perspective, I recount briefly my own journey in structural biology from a mass spectrometry (MS) perspective. I then highlight how MS techniques, especially cross-linking mass spectrometry (XL-MS), can be applied to fill in the missing pieces of structural information. XL-MS is a complementary technique that can help capture a more complete picture of the puzzle that is structural biology.

## A young scientist's journey in mass spectrometry and structural biology

MS proteomics has a wide range of applications, from quantifying protein abundance to identifying post-translational modifications. I started in MS proteomics before my PhD working on proximal biotinylated protein interactors using BioID. During my PhD, I worked on XL-MS on both *in situ* cross-linking and on the smc5/6 complex. As a post-doctoral scientist, I continued working on XL-MS on a wider number of protein complexes, ranging from dengue virus proteins to membrane proteins and antibody–antigen interactions. Each protein complex required different ways to analyze the XL-MS data together with other structural data, resulting in a different integrative biology approach each time. Drawing from my own experience, I share my perspective on how MS can contribute to structural biology, and how XL-MS complements other techniques by filling in some of the missing puzzle pieces of structural biology.

## Mass spectrometry in structural biology

Structural information from MS can help complement high-resolution protein structures (Pukala and Robinson, 2022). Such information includes stoichiometries of proteins in complex,

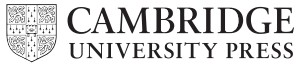

surface accessibility or reactivity of proteins, and distance constraints for protein residue-to-residue information. MS enables the measurement of the mass of biological molecules, and several techniques have been developed for structural biology applications. Native and top-down mass spectrometry enables measurement of the mass of protein complexes, enabling identification of protein components and their stoichiometries in protein complexes based on the mass of the proteins and proteins in the complex. Hydrogen-deuterium exchange mass spectrometry (HDX-MS) measures the rate of exchange of hydrogen to deuterium on proteins. This rate of exchange enables the inference of solvent exposure and determination of protein regions that are interacting or undergoing conformational changes. HDX-MS can be applied to map antibody epitopes and determine protein–protein interaction regions. Protein foot-printing methods identify protein residues that are susceptible to chemical modification. The susceptibility of chemical modification is used to infer the surface and solvent accessibility of protein residues. XL-MS uses chemical cross-linkers with defined spacer arm lengths to cross-link proteins and identify the cross-linked peptides by MS. Identification of residue-to-residue cross-links enables an upper distance limit to be assigned to each residue-to-residue interaction. This provides low-resolution structural information that can be used to infer protein-interacting regions and can be integrated with other structural information. More importantly, XL-MS can be performed on proteins in solution and provides an aggregate view of the protein structure and proximities of residues.

## Cross-linking mass spectrometry in structural biology

XL-MS has been used to great effect in structural biology. XL-MS has been successfully applied to large mega-Dalton protein complexes (Robinson *et al.*, 2016; Akey *et al.*, 2022), and has been used to map structures in cells when combined with AI-predicted

structures. (Bartolec *et al.*, 2023; O'Reilly *et al.*, 2023) XL-MS complements other structural biology techniques (Figure 1), especially in the following applications: identification of protein structure conformational changes; integration with other structural information for integrative structural modelling, and studying protein structures and interactions in tissue, in cells, or in situ.

## Cross-linked interactions

Protein structures are highly varied and different regions of proteins adopt different structures. Flexible regions of proteins, including structurally disordered regions and loop regions, are often not visualized by X-ray crystallography and cryo-EM. While structural modelling can help visualize these protein regions, the prediction of their interactions may not be accurate. XL-MS has the potential to map interactions of these flexible regions, based on experimental cross-linking data of these regions to proximal protein regions. My own work on the smc5/6 complex using XL-MS identifies interactions of the Nse5 HAD loop in the smc5/6 complex. The flexible loop region is not captured in the solved cryo-EM protein structure, but its interactions are captured by XL-MS (Yu *et al.*, 2022).

## Conformational changes and protein function

Protein structures are often presented as a discreet set of conformations which does not accurately represent their highly dynamic nature. Proteins adopt different structural conformations depending on their functional state, and engagement with ligands or substrates results in these conformational changes. XL-MS can be used to cross-link proteins in solution and in different conformations, such as in an ATP-binding state versus apo-state or in a DNA-bound state against a DNA-free state. Comparison of cross-links allows elucidation of structural changes associated with the different functional states. Combined with the potential to capture

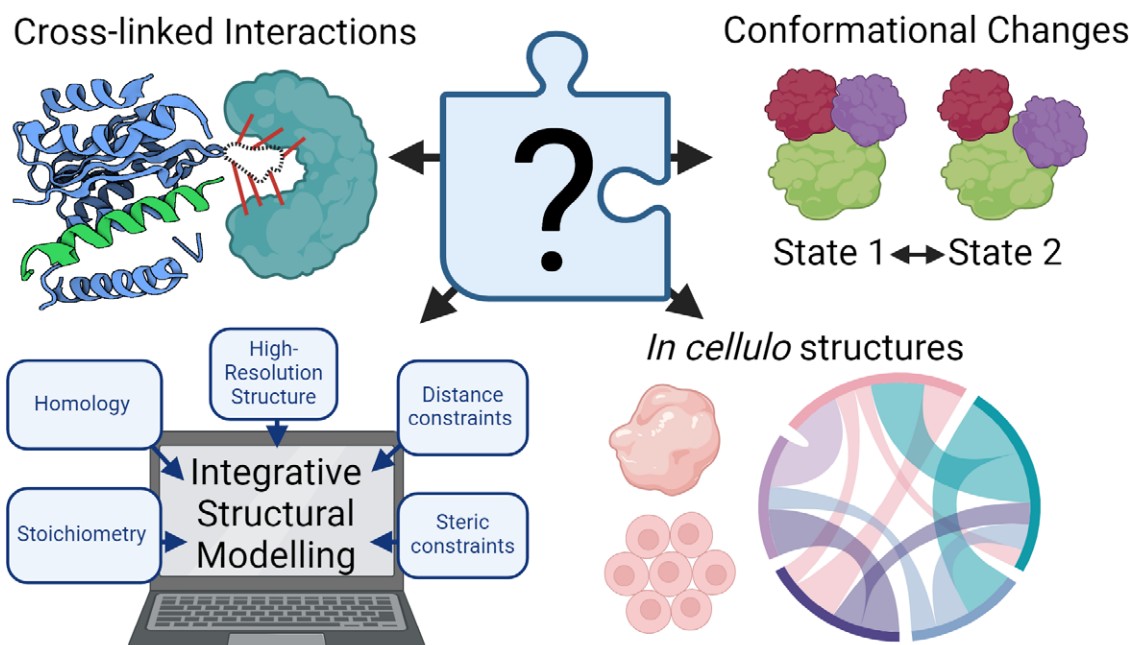

**Figure 1.** Cross-linking mass spectrometry can help fill in missing puzzle pieces in structural biology. Top left: Cross-linked interactions may capture structural interactions not identified by other structural techniques. Top right: Differential cross-linking patterns may be observed for different protein conformational states. Bottom left: Distance constraints from cross-linking mass spectrometry can be used together with other structural inputs for integrative structural modelling. Bottom right: Cross-linking mass spectrometry offers a unique look into protein interactions and structures *in cellulo*. (Image created with BioRender.com.)

interactions of flexible regions, XL-MS can provide a good snapshot of protein structural changes in solution from conformation to conformation.

Despite the application of XL-MS to identify conformational changes, there are some limitations to interpreting XL-MS data. First, XL-MS data typically captures the sum of all structural conformations in a given sample. A sample with a mix of protein structures will have cross-links reflecting the different mixes of protein structures that are not deconvoluted. This complicates data interpretation if the intent is to discern a particular structural conformation. Biasing the sample to a particular protein structure composition is recommended for clearer data interpretation and this can be achieved through purification of the protein complex or through the addition of excess ligand or substrate to 'lock' the protein complex in a particular state. Second, XL-MS usually informs of large conformational changes. Using commonly available cross-linkers such as disuccinimidyl sulfoxide (DSSO), cross-linked residues have an approximate upper limit C$\alpha$-C$\alpha$ distance of 20–30+ Å, which limits detection of structural changes. Conformational changes need to have the same order of magnitude or greater to be identifiable by XL-MS. Zero-length and short-length cross-linkers with spacer arms of $\leq 5$ Å, such as 1-ethyl-3-($-3$-dimethylaminopropyl) carbodiimide hydrochloride (EDC) and triazidotriazine (TATA), can provide better resolution for identification of structural changes. A combination of short- and long-length cross-linkers can be used to great effect to map structural changes to an even greater extent. (Brodie *et al.*, 2017) Additionally quantitative XL-MS, where cross-linked peptides are quantified, can improve the interpretation of structural changes from one protein state to another protein state.

### Integrative modelling of protein structures

Structural biologists have been integrating structural information from multiple techniques to refine our current protein structural models. These refined models more accurately reflect protein structure and function by integrating structural information from multiple sources and techniques. The refined structures can provide additional key insights for protein engineering or protein drug-targeting for biomedical purposes.

Different MS techniques are employed in different stages of integrative modelling to provide an array of structural information (Britt *et al.*, 2022). Native and top-down MS provides stoichiometries of the protein complex, limiting the number of proteins to be modelled. HDX-MS identifies protein regions that are undergoing conformational changes. Protein foot-printing methods identify residues that are solvent-exposed and susceptible to chemical modification. XL-MS provides distance restraints between specific residues, thus limiting the amount of conformational space needed to be sampled for modelling. XL-MS restraints can be applied to molecular dynamics (MD) simulations for conformational sampling, or to molecular docking to better model protein–protein interacting structures and protein complexes.

AI-based predictive structural modelling and its improved accuracy in predicting protein structures have resulted in the growth of AI-based structures. AI predicted structures are met with both great enthusiasm for their potential and with scepticism based on the limitations of AI-based approaches (Perrakis and Sixma, 2021). XL-MS provides interesting synergies with AI-based structure prediction. First, XL-MS can be an orthogonal experimental method to check the accuracy of AI-predicted structures. As XL-MS can be performed on proteins in solution with a short experimental turnaround time, a large number of cross-linked interactions can be obtained in a reasonable timeframe. The distance restraints from identified cross-links can then be used to gauge the accuracy of the AI-predicted structure. Second, XL-MS may be used to enhance AI-based predictive structural modelling. The distance restraints can be used together with the AI-predicted structure for molecular dynamics simulations or to improve the molecular docking of proteins in protein complexes. This may be especially important in areas where AI-based methods are known to perform poorly, such as antibody–antigen interactions. The use of XL-MS and other experimental data to complement the AI-predicted structures may improve the accuracy of the modelled structure.

### Protein structures *in cellulo*

XL-MS has been successfully applied to cells and tissue to capture *in cellulo* structural information of proteins and their complexes. This provides a unique advantage to XL-MS to identify structural information of proteins in their native context and allows comparison of *in cellulo* protein cross-links with protein structures from purified or recombinant proteins. Combining *in cellulo* XL-MS with AI-predicted protein structures provides a snapshot of the protein–protein interaction network with structural detail, or the structural interactome (Bartolec *et al.*, 2023; O'Reilly *et al.*, 2023).

### Towards a more complete picture of protein structures

Solving the puzzle that is protein structural biology can only be more effective with complementary tools and techniques. Here, I shared several ways in which XL-MS can act as a complement to existing structural biology techniques and approaches. Different proteins require different ways and applications of XL-MS and integrative structural approaches, highlighting the diverse and complex nature of protein structure. Further advancements in XL-MS will enable more comprehensive cross-link interactions to be acquired *in cellulo*, bridging the gap from *in vitro* structures to *in situ* structures. Applications of XL-MS to structure-based drug discovery are also likely to be developed. The use of multiple structural techniques in an integrative manner will help biologists capture a more complete picture of protein structures.

**Open peer review.** To view the open peer review materials for this article, please visit http://doi.org/10.1017/qrd.2024.13.

**Acknowledgements.** The author would like to thank Wint Wint Phoo for her helpful comments from reading the manuscript.

**Author contribution.** Z.S. conceptualized and wrote the manuscript.

**Financial support.** Z.S. was supported by an A*STAR Career Development Fund (Project ID: 212D800074) and an A*STAR Young Achiever Award.

**Competing interest.** The author declares no competing interest to declare.

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
