## [Reviewer Report]

In this article, Zheng reflects on the state of the field of structural biology and cross linking mass spectrometry, through the eyes of his own scientific journey and research experience. It reads well and covers the topic in question aptly. Publication of the manuscript is recommended.

Minor Comment

In page 1, first instance of mass spectrometry should be abbreviated already (A young scientist’s journey in mass spectrometry and structural biology. >>Mass spectrometry<< proteomics...). Abbreviation in the second page should be removed. Same issue with A.I. on page 3.

---

## [Reviewer Report]

The manuscript “Filling in missing puzzle pieces in protein structural biology with cross-linking mass spectrometry." by Dr. Ser Zheng summarizes a cross-linking mass spectrometry method and its applications. Overall, the manuscript is well written, and I believe it can be considered for publication after correcting the minor issues pointed out below.

1. The definition of abbreviations is not consistently formatted throughout the manuscript. For example, the term “cross-linking mass spectrometry” first appears in the first paragraph of the main body, but the abbreviation “XL-MS” is not defined at that point. It is defined later in the section “A young scientist’s journey in mass spectrometry and structural biology.”; however, the phrase “cross-linking mass spectrometry” is occasionally used instead of “XL-MS” after its definition. Additionally, "Artificial Intelligence (A.I.)” is introduced in the first paragraph, but is redefined in the section “Integrative modelling of protein structures”. Abbreviations should be defined upon their first appearance in the manuscript, and subsequent references should consistently use the abbreviated terms.

2. The author states, “Crosslinked residues have an approximate upper limit Cα-Cα distance of 20-30+ Å, which limits detection of structural changes” in the section “Conformational changes and protein function”. I largely agree with this statement, but in practice, researchers can select different lengths of cross-linkers to adjust this Cα-Cα length constraint. For example, Brodie et al. (10.1126/sciadv.1700479) reported using various lengths of cross-linkers to achieve higher resolution. It is important for readers to be aware that variations in the lengths of available cross-linking reagents exist.